

# A complete glacier inventory of the Antarctic Peninsula based on Landsat7 images from 2000-2002 and other pre-existing datasets

Jacqueline Huber[1], Alison J. Cook[2, 3], Frank Paul[1], Michael Zemp[1]

[1]Department of Geography, University of Zurich–Irchel, Zurich, 8057, CH
[2]Department of Geography, Swansea University, Swansea, SA2 SPP, UK
[3]Department of Geography, Durham University, Durham, DH1 3LE, UK

Correspondence to: M. Zemp (michael.zemp@geo.uzh.ch) and F. Paul (frank.paul@geo.uzh.ch)

**Abstract.** The glaciers on the Antarctic Peninsula (AP) potentially make a large contribution to sea level rise. However, this contribution has been difficult to estimate, as no complete glacier inventory (outlines, attributes, separation from the ice sheet) has been available. This work fills the gap and presents a new glacier inventory of the AP north of 70° S, based on digitally combining pre-existing datasets with GIS techniques. Rock outcrops have been removed from the glacier basin outlines of Cook et al. (2014) by digital intersection with the latest layer of

the Antarctic Digital Database (Burton-Johnson et al., 2016). Glacier-specific topographic parameters (e.g. mean elevation, slope and aspect) as well as hypsometry have been calculated from the Digital Elevation Model (DEM) of Cook et al. (2012). We also assigned connectivity levels to all glaciers following the concept by Rastner et al. (2012). Moreover, the bedrock dataset of Huss and Farinotti (2014) enabled us to add ice thickness and volume for each glacier.

The new inventory is available from the GLIMS database (doi: 10.7265/N5V98602) and consists of 1589 glaciers covering an area of 95 273 km$^2$, slightly more than the 90 000 km$^2$ covered by glaciers surrounding the Greenland Ice Sheet. The total ice volume is 34 590 km$^3$, of which 1/3 is below sea level. The hypsometric curve has a bimodal shape due to the unique topography of the AP, which consists mainly of ice caps with outlet glaciers. Most of the glacierized area is located at 200–500 m a.s.l. with a secondary maximum at 1500–1900 m. Approximately 63%

of the area is drained by marine-terminating glaciers, and ice shelf tributary glaciers cover 35% of the area. This combination indicates a high sensitivity of the glaciers to climate change for several reasons: (1) only slightly rising equilibrium line altitudes would expose huge additional areas to ablation, (2) rising ocean temperatures increase melting of marine terminating glaciers, and (3) ice shelves have a buttressing effect on their feeding glaciers and their collapse would alter glacier dynamics and strongly enhance ice loss (Rott et al., 2011). The new inventory

should facilitate modelling of the related effects using approaches tailored to glaciers for a more accurate determination of their future evolution and contribution to sea level rise.

## 1   Introduction

The ice masses of the Antarctic Peninsula (AP) potentially make a large contribution to sea level rise (SLR) as a large amount of water is stored in the ice and a high sensitivity to temperature increase has been reported (Hock et

al., 2009). However, the glaciers on the AP were not separately taken into account for their individual sea level contribution in the fifth Assessment Report of the IPCC (Vaughan et al., 2013) as a complete glacier inventory of the AP was not available at that time. As a result, only the ice masses of the surrounding islands have been considered from the inventory compiled by Bliss et al. (2013). The freely available datasets for the AP were incomplete and of a varied nature (see Figure 1), ranging from the World Glacier Inventory (WGI; WGMS and

NSIDC, 1989, updated 2012) that provides extended parameters for most of the glaciers on the AP from the second half of the 20[th] century but without area information and only available as point data, to the vector datasets (2D outlines) from the Global Land Ice Measurements from Space (GLIMS; GLIMS and NSIDC, 2005, updated 2015)



database and the Randolph Glacier inventory (RGI; Arendt et al., 2015) that were spatially incomplete. Moreover, the spatial overlap of the WGI with the boundaries of individual glaciers in the RGI was limited (Figure 1) so that

an automated digital intersection (spatial join) for parameter transfer was not possible.

Conversely, for Graham Land, representing the part of the AP north of 70° S, several more specific datasets exist that could be combined to a full and coherent glacier inventory: A detailed 100 m resolution digital elevation model (DEM) has been prepared by Cook et al. (2012); glacier catchment outlines based on this DEM and the Landsat Image Mosaic of Antarctica (LIMA; Bindschadler et al., 2008) were derived by Cook et al. (2014); a recently

updated dataset of rock outcrops for entire Antarctica is available from the Antarctic Digital Database (ADD; http://www.add.scar.org/home/add7), and a modelled raster dataset of bedrock topography is available from Huss and Farinotti (2014).

Here we present the first comprehensive glacier inventory of the Antarctic Peninsula north of 70° S (Graham Land) and describe methods used to digitally combine the existing datasets. The final outline dataset of the AP is

supplemented with several glacier-specific parameters, such as topographic information and hypsometry, thickness and volume information as well as the earlier classification of glacier front characteristics. With these parameters we analyse similarities and differences to other glacierized regions, as well as glacier-specific contributions to sea level and climate sensitivities. For a clear handling by different modelling and remote sensing communities, each glacier is assigned one of three connectivity levels to the ice sheet (CL0: no connection, CL1: weak connection,

and CL2: strong connection) following the approach introduced by Rastner et al. (2012) to separate the peripheral glaciers on Greenland from the ice sheet.

## 2   Study region

The AP extends northwards of the mainland from approximately 75° S for more than 1500 km north-easterly to 63° S, and is enclosed to the west by the Bellingshausen Sea and to the east by the Weddell Sea of the Southern

Ocean. The northern-most part of the AP from 70° S represents Graham Land and its peripheral islands, for which the glacier inventory is created. The South Shetland Islands are not regarded as being part of the AP and hence not included in the present inventory. The central part of the mainland is dominated by a narrow mountain chain with a mean height of 1500 m (maximum 3172 m) and an average width of 70 km. The unique topography, with an interior high-elevation plateau surrounded by steep slopes and flat valley bottoms results in distinct glacier types.

In general, the highest regions are covered by ice caps and much lower-lying valley glaciers are either connected to them and heavily crevassed in the steep regions, or they are entirely separated from them, uncovering several rock outcrops.

The AP has a polar to subpolar maritime climate, but the climatic and oceanographic regime varies across the AP causing varying glacier dynamics (Arigony-Neto et al., 2014). The often poly-thermal glaciers experience a distinct

melting period in summer, particularly the glaciers in the northern part of the AP. The special topographic characteristics of the AP make the flat, low-lying parts of its glaciers particularly vulnerable to climate change: For example, a small increase in temperature might cause large parts of their area to become ablation regions; most of them are marine-terminating glaciers that experience melt also from surrounding ocean waters (Cook et al., 2016), and many of them nourish ice shelves (Cook et al., 2014) that currently help buttress them but can quickly disappear

(Rott et al., 1996) causing rapid shrinkage of the related glaciers (Rott et al., 1996; Hulbe et al., 2008).



Since the early 1950s, significant atmospheric warming trends (Turner et al., 2009) and increasing ocean temperatures (Shepherd et al., 2003) have been observed across the AP. As a consequence, ice shelves are collapsing and glacier fronts are retreating (Pritchard and Vaughan, 2007; Davies et al., 2012; Cook et al., 2014 and 2016). On the other hand, knowledge about the mass balance of the glaciers of the AP is sparse (Rignot and

Thomas, 2002) although a few studies exist that indicate a general mass loss (Helm et al., 2014; Kunz et al., 2012). For the purpose of this study, the AP is additionally divided into four sectors (NW, NE, SW and SE) to reveal differences between climatically different regions of the AP. The division west/east is based on the main topographic divide, and north/south is based on the 66° S latitude.

## 3   Datasets

This Section gives a short description of the datasets covering the AP (Graham Land) that are used for generating the glacier inventory. Table 1 summarizes their key characteristics presenting their content, sources, access, references and application in this study. The following datasets are used:

1.  The digital elevation model (DEM) by Cook et al. (2012);
2.  the glacier catchment outlines by Cook et al. (2014);
3.  the rock outcrops dataset of Antarctica of Burton-Johnson et al. (2016);
4.  the bedrock elevation grid by Huss and Farinotti (2014);
5.  the Antarctic ice sheet drainage divides by Zwally et al. (2012); and
6.  Landsat Image Mosaic of Antarctica (LIMA) by Bindschadler et al. (2008)

### 3.1   Digital elevation model

Cook et al. (2012) generated a 100 m resolution DEM of the AP (63°–70° S), which is available from the National Snow and Ice Data Center (NSIDC; http://nsidc.org/data/NSIDC-0516) in the WGS84 Stereographic South Pole projection. This DEM is an improvement of the ASTER Global Digital Elevation Model (GDEM) product, which locally contained large errors and artifacts (cf. Cook et al., 2012). The accuracy of the DEM is in particular
improved on gentle slopes of the high plateau region. However, small anomalies were removed, which has resulted in small inherent gaps along the coast and some islands are missing (Cook et al., 2012). As a result, the DEM does not entirely cover the study region (approximately 1% of the area is missing). This DEM has also been used by Cook et al. (2014) for the generation of catchment outlines (see next Section) and is used in this study for the calculation of glacier specific parameters (see Sect. 4.3) for the glacierized areas it covers.

### 3.2   Catchment outlines

Glacier inventories, such as those available in GLIMS or the RGI, require glaciers to be separated into individual entities (Paul et al., 2009). This can be accomplished by digitally intersecting drainage divides derived from watershed analysis (e.g. Bolch et al., 2010; Kienholz et al., 2013) with outlines of glacier extents derived from semi-automated mapping techniques (e.g. Paul et al., 2002). Cook et al. (2014) automatically delineated glacier
catchments of the AP in ESRIs ArcGIS by applying hydrological tools to the DEM described above (Figure 2). The AP coastline and some islands in that dataset were digitized based on images acquired by Landsat 7 between



2000 and 2002 for the Landsat Image Mosaic of Antarctica (LIMA; Bindschadler et al., 2008). As the DEM misses some islands around the AP, mainly in the central western region, the drainage divide analysis is missing for these regions. Additionally, grounding lines from the Antarctic Surface Accumulation and Ice Discharge (ASAID)

project data source (Bindschadler et al., 2011) were used, modified in places with features visible on LIMA to divide glaciers from ice shelves. Furthermore, the ice velocity dataset of Rignot et al. (2011) was considered by Cook et al. (2014) to manually verify and adjust the lateral boundaries of glaciers.

The resulting dataset consists of 1590 glacier catchment outlines for the AP with an area of 96 982 km², covering the region between 63°–70° S. Islands smaller than 0.5 km² and ice shelves are excluded. The dataset provides a

consistent time period of all basins and includes several parameters for each basin such as location, time stamp, area, and a classification of glacier type, form and front. The definition of the parameters and category numbers conform to the GLIMS classification system provided by the GLIMS Classification Manual (Rau et al., 2005) and based on the UNESCO (1970) guidelines as well as the Glossary of Glacier Mass Balance (Cogley et al., 2011). However, topographic parameters such as minimum, maximum, mean, and median elevation, or mean slope and

aspect are missing.

This catchment outlines dataset is available from the Scientific Committee on Antarctic Research (SCAR) Antarctic Digital Database (ADD; http://add.scar.org/home/add7; ADD Consortium, 2012), but does not include any of the glacier-specific attributes mentioned above besides area and length. The dataset with the complete information has not been published so far and has been provided directly by A. Cook for the purpose of this project in the WGS84

Stereographic South Pole projection. Whereas the catchment outlines provide a solid foundation for the generation of a glacier inventory, rock outcrops are part of the glacierized area and need to be removed (Raup and Khalsa, 2010).

### 3.3   Rock outcrops

The ADD website (www.add.scar.org) provides a detailed vector dataset of rock outcrop boundaries in the WGS84

Stereographic South Pole projection that has recently been updated (cf. Burton-Johnson et al., 2016). A former rock outcrops dataset, which has already been used for instance by Bliss et al. (2013) to create the inventory for the glaciers of the islands surrounding Antarctic, originated from a digitization of outcrops from different maps prepared in the 1990s at different scales and with variable accuracy. As a result, the dataset has some major georeferencing inconsistencies, misclassifications and overestimation of the ice-free area of Antarctica (Burton-

Johnson et al., 2016). The recently improved dataset of exposed rock outcrops by Burton-Johnson et al. (2016) used here (Figure 2), overcomes these issues and has a much better accuracy. It is based on a new automated method that identifies sunlit as well as shaded rock outcrops using multispectral classification of Landsat 8 satellite imagery. Incorrectly classified pixels (illuminated and shaded) such as snow, clouds and liquid water have been removed manually. The new dataset reveals that 0.18% of the total area of Antarctica are rock outcrops, which is

approximately one half of previous estimates (Burton-Johnson et al., 2016).

### 3.4   Bedrock elevation grid

Huss and Farinotti (2014) derived a new bedrock elevation grid with 100 m spatial resolution as well as the related ice thickness grid based on glacier surface topography and simple ice dynamic modelling. Compared to the



Bedmap2 dataset by Fretwell et al. (2013) with a resolution of 1 km, the new version also captures the rugged

subglacial topography in great detail. The narrow and deep subglacial valleys that are often below sea level are more accurately represented, allowing the modelling of even small-scale processes.

Their dataset is available online from the article supplement (doi: 10.5194/tc-8-1261-2014-supplement) on WGS84 Antarctic Polar Stereographic projection. Their dataset already excluded the rock outcrops using the former version of the ADD (ADD Consortium, 2012). As we have used the updated version of the rock outcrops dataset for creating

the glacier inventory, a new thickness grid is calculated (see Sect. 4.1).

### 3.5    Antarctic ice sheet drainage divides

The Cryosphere Science Laboratory of NASA's Earth Sciences Divisions (Zwally et al., 2012) provide an Antarctic ice sheet drainage divides dataset developed by the Goddard Ice Altimetry Group from ICESat data based on the GLAS/ICESat 500 m laser altimetry DEM (DiMarzio, 2007). Other sources, such as the Landsat Image Mosaic of

Antarctica (Bindschadler et al., 2008) and the MODIS Mosaic of Antarctica (Haran et al., 2005), were used as a guide to refine the drainage divides. Ice sheet drainage systems were delineated to identify regions broadly homogeneous regarding surface slope orientation relative to atmospheric advection and denoting the ice sheet areas feeding large ice shelves. The AP is assigned to four different basins (drainage system ID 24-27) with a relatively clear separation from the ice sheet along 70º S latitude (see Sect. 4.2).

## 4    Methods

The data generation workflow is roughly divided into four steps: (1) Intersecting datasets, (2) defining connectivity levels, (3) calculating glacier-specific attributes (topographic parameters), including ice thickness and volume information, and (4) the calculation of the overall and the glacier-specific hypsometry. All calculations are performed with various tools available in ESRI's ArcGIS version 10.2.2. All of the functionality is also available

in other GIS software packages. The four main steps are described in the following in more detail.

### 4.1    Intersecting datasets

When generating an inventory based on the semi-automated band ratio method (Paul et al., 2009), rock outcrops are automatically excluded from the glacier area. In this study the glacier catchment outlines are digitally intersected with the latest vector dataset of rock outcrop boundaries from the ADD (see Sect. 3.3). By removing

the new rock outcrops from the catchment outlines of Cook et al. (2014) a mask of individual glaciers is generated, assuming that areas not identified as rock outcrops are ice covered. Apart from the rock outcrops, the dataset of Cook et al. (2014) is generally in agreement with the procedures and GLIMS guidelines (Racoviteanu et al., 2009; Raup and Khalsa, 2010) for deriving glacier information.

To include glacier specific ice thickness and volume information, the bedrock grid of Huss and Farinotti (2014) is

subtracted from the DEM of Cook et al. (2012) and combined with the new glacier outlines. A grid with ice volume is then derived by multiplying the ice thickness grid with the cell area (10 000 m$^2$).



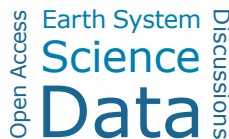

### 4.2 Defining connectivity levels

Rastner et al. (2012) suggested that peripheral glaciers on Greenland with a strong connection with respect to ice dynamics to the Greenland ice sheet should be regarded as part of the ice sheet and assigned the connectivity level 190  2 (CL2). This is where glaciers have a strong connection to the ice sheet and the location of their drainage divide on the DEM is uncertain due to the low-sloping terrain. For the Antarctic ice sheet drainage divides (see Sect. 3.5), basins south of 70° S are strongly connected to the West Antarctic ice sheet. Accordingly, they are assigned CL2 and are not included or further considered in the inventory presented here. The assignment of CL1 (i.e. weak connectivity to ice sheet) to the glaciers on the mainland and north of 70° S is performed automatically within the 195  GIS following the heritage rule introduced by Rastner et al. (2012), i.e. a glacier connected to a glacier assigned CL1 will also receive the attribute CL1. With this strategy, all glaciers on surrounding islands (i.e. those in the inventory from Bliss et al. (2013)) are assigned the value CL0. Large glaciers that are theoretically separable but otherwise closely connected to the ice sheet (e.g. Pine Island and Thwaites) have the value CL2.

### 4.3 Glacier-specific topographic parameters, ice thickness and volume

All glacier specific attributes (minimum, maximum, mean, and median elevation, mean slope, aspect, and thickness, total ice volume and ice volume grounded below sea level) are calculated by digitally combining the glacier outlines with the DEM, the ice thickness and volume grids using the zonal statistics tool in ArcGIS. This tool statistically summarizes the values of the underlying raster datasets (e.g. DEM, ice thickness) within specific zones with a unique ID (glacier outlines) and writes the results to an attribute table. The table is joined with the 205  attribute table of the glacier outlines dataset based on a common and unique identifier in both tables (i.e. the Glacier ID). All calculations are performed using the WGS84 South Pole Lambert Azimuthal Equal Area projection.

As the bedrock and hence also thickness datasets are based, inter alia, on the DEM of Cook et al. (2012) they are not universally spatially congruent with the glacier outlines (i.e. the boundary limits differ between the outlines and the other datasets). Of the 1589 glacier outlines, the thickness and volume values could not be calculated for 210  50 glaciers of the inventory. Accordingly, the topographic parameters, thickness and volume values of the glaciers on the islands that are not completely covered by the ice thickness and bedrock dataset do not represent values for complete glaciers. In addition, two glaciers are insufficiently covered by the 100x100 m pixel of the DEM. Hence these glaciers are not or insufficiently covered by the bedrock dataset of Huss and Farinotti (2014). Hence, 1541 glaciers have topographic information and 1539 glaciers have thickness, volume and SLE information, of which 215  some only have partial ice thickness and volume information.

To estimate the volume grounded below sea level for each glacier, a grid representing the distribution of the volume grounded below sea level is calculated by extracting the areas of the bedrock grid with negative values (areas below sea level).

The sea level equivalent (SLE) of the ice volume is calculated by first multiplying the volume of ice by 0.9 220  (assuming a mean ice density of 900 kg m$^{-3}$ not taking into account firn-air content) to determine the corresponding volume of water. Dividing the volume of water by the ocean surface area ($3.625 \times 10^8$ km$^2$; Cogley, 2012) gives the sea level contribution of the volume in m SLE, assuming all ice volume contributes to sea level if melted. This is not the case for the grounded ice below sea level, which has a negative (lowering) effect as this volume will be replaced by water with a higher density (Cogley et al., 2011). This effect has been considered in a second step in



the presented SLE estimations. Other effects, such as the cooling and dilution effect on ocean waters by floating
ice (Jenkins and Holland, 2007), are not taken into account here.

### 4.4 Glacier hypsometry

The distribution of the glacierized area with elevation (hypsometry) is calculated (a) for the entire AP in 100 m
elevation bins, (b) for the four sub-regions also in 100 m bins, and (c) for each individual glacier using 50 m
elevation bins. The calculation is based on the DEM of Cook et al. (2012) that is converted to 100 bins using the
'reclassify' tool and the 'extract by mask' tool for the respective sub-regions. Additionally, the hypsometry of the
catchment outlines is calculated to determine the effect of removing rock outcrops on the hypsometry. For further
comparisons we also calculated the hypsometry of the marine and ice shelf terminating glaciers and the hypsometry
of the bedrock.

## 235 5 Results

### 5.1 Size distribution

The glacier inventory for the AP ranges from 63°–70° S to 55°–70° W and consists of 1589 glaciers covering an
area of 95 273.2 km$^2$ (Figure 3a) without rock outcrops, ice shelves and islands <0.5 km$^2$. The exclusion of the rock
outcrops from the earlier catchment outlines reduced the total area by 1709.4 km$^2$. The smallest and largest glaciers
have an area of 0.3 km$^2$ and 7018.4km$^2$, respectively. The 619 glaciers located on islands (CL0) cover an area of
14 299 km$^2$ representing 15% of the total glacierized area. The remaining 970 glaciers are located on the mainland
(CL1) covering 80 974 km$^2$ and hence 85% of the total area. As the DEM is spatially not perfectly congruent with
the glacier outlines, of the total 1589 glacier outlines 48 outlines do not have any elevation information. As a result,
the calculations including the DEM, the bedrock or thickness dataset are applied only to 1541 glaciers, of which
some only have partial elevation information. In Table 2 all parameters of the attribute table are listed, including
the corresponding values of an example glacier (for location see inset map in Figure 3a). The hypsometry of each
individual glacier, as exemplified in Figure 3b, is stored and available separately in an csv-file. Several parameters,
such as primary classification, glacier form and front, and meta-data about the satellite image have been determined
and provided by Cook et al. (2014), as defined for the GLIMS inventory. Others (i.e. connectivity levels,
topographic parameters, ice thickness and volume) are the result of the calculations described in Sect. 4. The
inventory is available for download from the GLIMS website: http://www.glims.org/maps/glims (doi:
10.7265/N5V98602).

Regarding the connectivity levels, all glaciers on islands surrounding the AP are assigned CL0 (no connection) and
the glaciers on the mainland are all assigned CL1 (weak connection). Even the glaciers at the very northern part of
the AP have CL1, due to the applied topological heritage rule (a glacier connected to a glacier assigned CL1 also
receives CL1). As the glaciers further south are in connection with the ice sheet, they have CL2 (strong connection),
are regarded as part of the ice sheet and hence are not included in the present dataset.
Figure 4a portrays the percentages per size class in terms of number and area. The mean area (59.96 km$^2$) is
considerably higher than the median area (8.27 km$^2$), reflecting the areal dominance of a few larger glaciers. Most



of the glaciers can be found in the size classes 4–6 (1.0–50 km²). These glaciers account for 77% of the total number but only for 14% of the total area. The glaciers larger than 100 km² cover the majority of the area (77%) yet comprise only 11% of the total number. With an area of 7018 km² Seller Glacier is the largest, accounting for 7% of the total area and being twice as large as the second largest glacier (Mercator Ice Piedmont, 3499 km²).

## 5.2   Topographic parameters

Figure 5a and b present a scatter plot of area against mean/median and area against minimum/maximum elevation, revealing that mean, median and maximum elevation is increasing towards larger glaciers. Three glaciers have a maximum elevation above 3100 m a.s.l., being 300 m or higher than all other glaciers. The highest elevation is in the south with 3172 m. Many glaciers have a minimum elevation of (close to) 0 m a.s.l. as most of them are marine-terminating. The average mean elevation of the 1541 glaciers with elevation information is 409 m and their median elevation is 317 m a.s.l. The spatial distribution of median elevation reveals an increase from the coast and islands (0–500 m a.s.l.) to the interior of the AP (up to about 1800 m a.s.l.). This can be seen in Supplement 1.

Figure 4b shows the distribution of glacier number and area as percentages of the total for each aspect sector of the AP. The distribution is rather balanced and is not revealing any trends. Somewhat fewer glaciers and areas have aspects from south to south-east. The large value in area of the south-western sector derives from the contribution of the largest glacier of the region (Seller Glacier). When mean aspect is plotted against mean elevation (Figure 6) there are also no significant trends. However, the highest mean elevation values are lower in the south-eastern sector. The scatter plot of mean slope against area (Supplement 2) reveals the common dependence on glacier size where mean slope decreases towards larger glaciers. Additionally, the scatter is smaller the larger the glacier, indicating that small glaciers exhibit a larger range of slope inclination.

The mean thickness of all 1539 glaciers involving thickness information is 130 m. The Eureka glacier, located in the south, has the largest mean thickness of all CL0 and CL1 glaciers with 851 m. The dependence of mean thickness on area and slope (indicating that the steeper/smaller the glacier, the thinner the ice) (Figure 7a and b), is not surprising, as ice thickness is modelled based on surface topography (Huss and Farinotti, 2012 and 2014). However, low-sloping glaciers reveal a large range of mean thickness values. The large but low-sloping glaciers of the high plateau and those in the very south towards the Antarctic ice sheet form a cluster of glaciers with higher mean thicknesses. The many small glaciers along the coast are mostly thin. The mean thicknesses per sector and per mean aspect (Supplement 3) do not reveal any significant spatial patterns.

The total ice volume of the AP is 33 770 km³. As the volume is calculated based on the thickness dataset, the volume distribution is basically a reflection of the thickness distribution. Table 3 lists the total volume per sector, revealing that most of the ice volume can be found in the south-west and south-east sector (38.6% and 32% of the total). This is not surprising as these two sectors make up 63% of the total glacierized area. Regarding the glacier volume per glacier area for individual glaciers, the highest values are found for the large glaciers at the very south of the AP, adjacent to the ice masses regarded as being a part of the Antarctic ice sheet.



Numerous, partly very pronounced, valleys lie below sea level especially in the north-eastern sector (the bedrock lying below sea level is visualised in Supplement 4). In total, approximately one third of the total grounded ice

volume is below sea level (Table 3), which has a negative effect on SLR (sea level lowering). About 50% of the volume of the north-eastern sector is grounded below sea level (Table 3). Although the negative effect on SLR is very small, this effect can now be better considered for future sea level estimations. Based on the results presented here, this results in a total estimated SLE of 54 mm (Table 3).

As mentioned before, the nominal glacier parameters primary classification, glacier from and front, have been determined by and described in Cook et al. (2014). They further illustrate the number of glaciers within each classification and frontal-type, which is therefore not repeated here.

### 5.3 Hypsometry

Figures 8a and b depict the glacier hypsometry (area-altitude distribution) for (a) the entire AP and for (b) each sector, revealing a bimodal shape of the hypsometry. Figure 8a additionally displays the effect of removing the rock outcrops from the outlines, the hypsometry only for marine-terminating and ice shelf nourishing glaciers as well as the hypsometry of the underlying bedrock. Exclusion of the rock outcrops, with a total area of 1709.4 km$^2$, does not change the general shape of the hypsometry. However, it slightly reduces the glacierized areas below 1500

m a.s.l. with a maximum areal reduction at 200–600 m and 1000–1200 m a.s.l. The hypsometry for marine-terminating and ice shelf nourishing glaciers confirms that most of the glacierized area is covered by these types. Additionally, these types are extending over the entire elevation range. Accordingly, the bimodal shape of the curve does not arise from different glacier (types) at lower and higher elevations. Rather, it is determined by and reflects the topography of the AP: The low-sloping and low-lying coast regions covered by valley glaciers account for the

maximum of the glacierized area between approximately 200–500 m a.s.l. The glacierized plateau region accounts for a secondary maximum at about 1500–1900 m a.s.l. The steep valley walls connecting the plateau with the coastal region result in the minimum at about 800–1400 m a.s.l. In addition, the hypsometry reveals that approximately 6000 km$^2$ of the 93 767 km$^2$ glacierized area covered by the DEM are found in the lowest elevation band (0–100 m). These areas are in direct or in close contact with water or ice shelves


The hypsometry per AP sector (Figure 8b; all excluding rock outcrops), reveals that in the two northern sectors both maxima of the hypsometric curve are less than those of to the two southern sectors. The elevations of the maxima are about the same for NW, NE and SW, whereas both maxima of the SE sector are somewhat lower. The glacier cover per sector reflects the bedrock topography of each sector. The bedrock of the northern sectors has less

area in the high plateau regions and therefore most of the glacierized areas are at lower elevations. The southern sectors have a more dominant plateau region favoring more glacierized areas at higher elevations compared to the northern sectors. However, the north-eastern sector has the largest fraction of glacierized area in the lowest 100 m and therefore in direct or in close water or ice shelf contact.



## 6    Discussion

### 6.1    Source data

This study has presented a complete and now publicly available glacier inventory for the Antarctic Peninsula north of 70° S that has been compiled from the best and most recent pre-existing datasets, complemented with information for individual glaciers that was not available before (topographic parameters, hypsography and ice thickness). To allow traceability of source data, we have not altered or corrected the available datasets despite some obvious shortcomings. For example, the DEM by Cook et al. (2012) does not cover all glaciers and several only partly, but we have not attempted to fill these missing regions with other source data (e.g. the ASTER GDEM). Consequently, the sample of glaciers with complete attribute information (1539) is reduced compared to the number of all glaciers in the study region (1589). The same applies for glaciers with a modelled ice thickness distribution. For 48 of these glaciers the DEM information was incomplete and ice thickness was accordingly not modelled by Huss and Farinotti (2014). Similarly, for rock outcrops: although the reported accuracy is only 85±8% and we could identify wrongly classified rock outcrops in comparison to LIMA, we used them as they are. This helps to also be consistent with other studies that will use the same datasets for their purpose. For the same reasons (consistency, traceability) we have also not corrected basin outlines or drainage divides using flow velocity fields derived from satellite sensors, also because this has already been done by Cook et al. (2014) for the catchment outlines. Alterations here would also impact on the already existing detailed classification of glacier fronts and we think it is better not to change this at this stage. Overall, results are as good as the source data used and their errors or incompleteness fully propagates into the products we have created here. However, we do not expect any major changes of the glacier characteristics or our overall conclusions with such corrections being implemented. On the other hand, addressing the shortcomings and improving the related datasets is certainly an issue to be considered for future work.

### 6.2    Comparison with other regions

In comparison with other recently compiled glacier inventories in regions of similar environmental conditions (mountainous coastal regions with maritime climate) such as Alaska (Kienholz et al., 2015), Greenland (Rastner et al., 2012) and Svalbard (Nuth et al., 2013), the AP has the largest glacierized area (94 743 km$^2$), closely followed by Greenland (89 720 km$^2$), Alaska (86 723 km$^2$) and with some distance by Svalbard (33 775 km$^2$). The AP has also the largest absolute, although only the second largest relative, area covered by marine-terminating glaciers, which are expected to react very sensitively to small changes of climate and associated ocean temperature changes. The glacier number and area distributions in the corresponding studies of Alaska, Greenland, Svalbard and the AP reveal that a few larger glaciers contribute most to the area in all regions. This dominance is also reflected in a median area, which is considerably smaller than the mean area. However, in Alaska, Greenland and Svalbard the number of small glaciers is distinctively higher, with maximum counts between 0.25 and 1 km$^2$. The glaciers of the AP do not exhibit this pattern, which confirms findings by Pfeffer et al. (2014) for glaciers in the RGI region Antarctic and Subantarctic. Only the glaciers on Svalbard have a favoured northern aspect (Nuth et al., 2013), which is interpreted as an evidence for the importance of solar radiation incidence for glacier distribution in this region (Evans and Cox, 2010).

Very different is the bimodal hypsometric curve for the glaciers on the AP (Figure 8) compared to the parabolic shape of the three other regions that have increasing area percentages towards their mid-elevation. Hence, the AP





has most of its glacierized area at lower elevations (around 200–500 m) with a secondary peak at higher elevations (around 1500–1900 m). As the hypsometry of a glacier is an indicator of its climatic sensitivity (Jiskoot et al., 2009) this comparison reveals that the future evolution of the AP glaciers cannot be modelled with the same simplified approaches as developed for glaciers in other regions (Raper et al., 2000) and that volume loss for a small rise in ELA might indeed be high (Hock et al., 2009). The aspect preference with poleward tendencies of glacier distribution that is common in other mountain ranges (Evans, 2006; Evans and Cox, 2005; Evans, 2007; Evans and Cox, 2010) could not be found for the AP as the entire AP is glacierized and most glaciers are marine terminating.

### 6.3    Uncertainties

#### 6.3.1    DEM

The DEM of Cook et al. (2012) provides currently the best resolution and covers the area of the AP most accurately. However, the DEM only covers 93 250 km$^2$ and hence 98.4% of the total glacierized area. Therefore, the calculation of the 3D parameters (mean, median, min., max. elevation, slope, aspect) was not possible for 48 glaciers, which are entirely excluded by the DEM, representing an area of 1044 km$^2$, about 3% of the total number and 1% of the total area. Also, some glaciers are only partially covered by the DEM, for instance one glacier is only covered by one pixel of the DEM, the values of their 3D parameters are based only on a restricted area and are not representative for the entire glacier. The problem is illustrated in Supplement 5. The region of Renaud and Biscoe Islands at the mid-western coast of the AP exemplifies that some outlines do not have any elevation information, and others do have elevation information but not for the entire glacier outline.

Even though only about 1.6% of the total glacierized area is not covered by the DEM and it is mainly a few islands that are affected, their values of the 3D parameters are expected to have slight inaccuracies. In addition, as the glacier hypsometry is calculated based on the DEM, the curves represent only the area covered by the DEM. Therefore, these curves might also be slightly altered by including the missing areas. Furthermore, as the ice thickness and bedrock dataset of Huss and Farinotti (2014) is based on the DEM of Cook et al. (2012), the mean thickness and the volume could not be calculated for 50 glaciers. Accordingly, the values of mean thickness and the total volume for the glaciers on the islands, which are not completely covered by the ice thickness and bedrock dataset are not representative for the entire glaciers. As a fast and result oriented solution, a new DEM could be generated by completing the incomplete outlined areas based on statistics of the part of the area including elevation information. For instance, a glacier with missing values at the coast the DEM would be filled with the lowest occurring elevation value. This would counteract the problem of glaciers with incomplete elevation information. However, the accuracy and hence adequacy of this method would have to be further assessed. In addition, it does not solve the problem for areas were no elevation information is available at all. Hence, another DEM should be used or a new DEM should be generated to provide elevation information for the remaining 48 glaciers. Cook et al. (2012) gives a overview on high resolution elevation datasets for the AP and how existing DEMs can be improved. Accordingly, the bedrock and thickness dataset should be completed for the areas of missing information, for instance, based on the procedures of Huss and Farinotti (2014).



### 6.3.2 Rock outcrops

The new rock outcrops dataset already achieved much higher and more consistent accuracies than the former dataset provided by the ADD. Nevertheless, the mean value for correct pixel identification is 85 ± 8% (Burton-Johnson et al., 2016). Hence, some areas are still misclassified. These uncertainties also influence the accuracy of the ice thickness grid and the resulting values as the thickness grid is also intersected with the rock outcrops. Misclassifications are partially detectable by overlaying the rock outcrops over the LIMA. However, time-consuming manual correction of these errors in the entire region would prohibit the traceability that is important to maintain. Hence, future improvements of a rock outcrops dataset could improve the accuracy of the inventory.

### 6.3.3 Grounding line

To divide the glaciers from ice shelves, Cook et al. (2014) used the grounding line based on the Antarctic Surface Accumulation and Ice Discharge (ASAID) project data source (Bindschadler et al., 2011), modified in places with features visible on the LIMA. The definition of the location of the grounding line significantly influences the extent of a glacier, and although these locations fluctuate over time, the data represented the most accurate positions corresponding with the times of the glacier front positions.

### 6.4 Assignment of connectivity levels

In the south, the assignment of connectivity levels corresponds with the Antarctic ice sheet drainage divides from the Cryosphere Science Laboratory of NASA's Earth Sciences Divisions (Zwally et al., 2012), which has assigned all unconnected glaciers (on islands) a CL0 and all glaciers on the AP CL1, following the suggestion by Rastner et al. (2012) for peripheral glaciers on Greenland. It is certainly the simplest possibility for such an assignment, but we think it is nevertheless sensible and fulfils its purpose. Consistency with earlier applications (e.g. all glaciers in the inventory by Bliss et al. (2013) have CL0) and transparency of the method are further benefits. It also allows the glacier and ice sheet measuring and modelling communities to perform their work with their respective methods and determine, for example, past or future mass loss / sea level contributions independently. This would allow a cross check of methods for individual glaciers that are not resolved (such as results from gravimetry and glacier models) and possibly also explain remaining differences between methods (Shepherd et al., 2012; Briggs et al., subm.). The problem of double-counting the contributions can also be avoided.

### 6.5 Relevance for future studies

The equilibrium line altitude (ELA) for a balanced budget (ELA0) can be well approximated from topographic indices such as the mean, median or mid-point elevation of a glacier (e.g. Braithwaite and Raper, 2009). Its value is not only a good proxy for precipitation (Ohmura, 1992; Oerlemans, 2005) but also very useful to model the effect of rising temperatures on future glacier extent (Paul et al., 2007; Cogley et al., 2011, 2011; Zemp et al., 2006; Zemp et al., 2007). However, as the glaciers of the AP, similar to the glaciers of Svalbard, are mainly marine-terminating glaciers (Vaughan et al., 2013), the lower limit of the glacier is predefined and the above proxies do not work as an approximation of the ELA as the variability of these proxies is only determined by the variability of the topography (i.e. its maximum elevation). The increasing median elevation towards the interior (Supplement 1) does not come from decreasing precipitation towards the interior but is an artefact of glacier distribution and depends





on whether a glacier reaches sea level or not. The analysis of the median elevation of the AP's land terminating glaciers is not satisfying as only nine glaciers are classified as land terminating glaciers. Hence, a different approach

has to be applied to analyse the precipitation pattern on the AP.

The bimodal shape of the hypsometry, revealing half of the glacierized areas being situated below 800 m a.s.l., as well as due to the high areal fraction of marine terminating and ice shelf tributary glaciers, raises the expectation of high glacier sensitivities to future climate changes, associated with rising air and water temperatures. With

respect to the hypsometry, the sensitivity is expected to be higher for the AP, compared to regions such as Alaska, Greenland and Svalbard. Depending on the current ELA on the AP, it is highly likely that a rising ELA will cause additional glacierized areas to be in the ablation zone. Regarding the sensitivity to changes in ocean temperatures, the sensitivity of glaciers on the AP is higher than other glacierized regions due to the high areal fraction of marine-terminating glaciers. Hence, future studies should further investigate but also consider the sensitivity of these

glaciers to ocean temperature changes.

The total ice volume as well as the volume below sea level is necessary for accurate estimations of the sea level contribution. The approximate calculation given here demonstrates the high relevance of this region regarding future sea level rise. In addition, the region is highly sensitive due to its bimodal area distribution with large parts

being located at very low elevations (see Figure 8). At 54 mm the AP's glaciers have a much higher contribution potential than the glaciers of Alaska (45 mm), Central Asia (10 mm), Greenland periphery (38 mm), Russian Arctic (31 mm) or Svalbard (20 mm). In total, the global glaciers have a potential SLR of approximately 374 mm (Huss and Hock, 2015) to 500 mm (Huss and Farinotti, 2012; Vaughan et al., 2013; Paul, 2011), which is still significant for low-lying coastal regions (Paul, 2011). Compared to the Antarctic ice sheet, with a SLE of 58.3 m (Vaughan et

al., 2013), the SLE of the AP seems negligible. However, regarding the high sensitivity and short response times of these glaciers on climate change, they are expected to be major contributors to SLR in the next decades. Moreover, the contribution of the AP's glaciers has not yet been fully considered in most of the studies. The new inventory can now be used to model the evolution of these glaciers explicitly with the current best approaches (e.g. Huss and Hock, 2015). The results presented here allow an approximation of the consequences of on-going and

future climate change for this region: The lowest 800 m and hence 50% of the glacierized area are prone to rising ablation and mass loss, causing a sea level contribution of roughly 50% of the total AP SLE (27 mm). Regarding the glacier termini, about 30% of the glacierized areas are attached to the Larsen C ice shelf. Collapse of this ice shelf (similar to Larsen A and B), which may happen due to a growing rift (Jansen et al., 2015), would cause rapid dynamic thinning of its tributary glaciers (e.g. Rott et al., 2011) due to the loss of buttressing. Hence, about 15%

of the SLE or approximately 9 mm, again considering the lowest 800 m, are indirectly attached to the Larsen C.

## 7    Conclusions

The compilation of a glacier inventory of the AP (63°–70° S, Graham Land), consisting of glacier outlines accompanied with glacier specific parameters, was achieved by combining already existing datasets with GIS techniques. The exclusion of rock outcrops by using the latest corresponding dataset of the ADD (Burton-Johnson

et al., 2016) from the glacier catchment outlines of Cook et al. (2014) resulted in 1589 glacier outlines (excluding





ice shelves and islands <0.05 km²), covering an area of 95 273 km². Combining the outlines with the DEM of Cook et al. (2012) enabled us to derive several topographic parameters for each glacier. By applying the bedrock dataset of Huss and Farinotti (2014), volume and mean thickness information was calculated for each glacier.

Connectivity levels with the ice sheet were assigned to all glaciers following Rastner et al. (2012) to facilitate observations / modelling by different groups. We started with a simple and transparent rule: Glaciers south of 70° S (Palmer Land) have CL2 and are regarded as being part of the ice sheet, all glaciers north of it and on the AP have CL1 and all glaciers on surrounding islands have CL0. The resulting inventory as well as its quality is largely influenced by the availability and accessibility of accurate auxiliary datasets. For instance, the DEM does not entirely cover the glacierized area, but 98.4%. Hence, for fewer than 50 glaciers the topographic glacier parameters as well as thickness and volume information are missing. For other glaciers these values are not representative for the entire glacier extent as smaller parts are missing. Future improved DEMs might help solve this problem.

As GLIMS now provides the complete glacier outlines dataset of the AP (see glims.org), a significant gap in the global glacier inventory has been closed and a major contribution for forthcoming regional and global glaciological investigations can be made. Furthermore, this demonstrates the potential of inventory data for improving the knowledge about glacier characteristics, sensitivities, similarities and differences to other glacier regions. With the full inventory now being freely available, approaches to improve, extend and further investigate the glaciers of the AP are encouraged. This will provide new insights about the glaciers of the AP, their behaviour in response to a changing climate and corresponding contribution to SLR.

**Author contribution**

J. Huber compiled and analyzed this dataset in the framework of a Master's thesis and prepared the manuscript.

F. Paul and M. Zemp supervised the thesis and helped with the preparation of the manuscript.

A. Cook generated the glacier catchment dataset and provided this dataset including information about the generation process.

**Acknowledgments**

Jacqueline Huber and Michael Zemp acknowledge financial support by the Swiss GCOS at the Federal Office of Meteorology and Climatology MeteoSwiss. The work by Frank Paul is funded by the ESA Project Glaciers_cci (4000109873/14/I–NB). We are grateful to the LIMA Project, the Antarctic Digital Database, the National Snow and Ice Data Center and The Cryosphere Science Laboratory of NASA's Earth Sciences Divisions for free download of their data, allowing us to realise this study. We would like to thank Matthias Huss and Daniel Farinotti for making their data available as well as for their practical input. The inventory and the study have benefited greatly from their provided bedrock dataset. Finally, we would like to thank R. Drews for the constructive suggestions to improve the quality of this paper.



## Tables

**Table 1. Datasets used for the generation of the glacier inventory and a description of their properties.**

|  | DEM | Glacier catchment outlines | Rock outcrops | Bedrock elevation grid | Antarctic ice sheet drainage divides |
|---|---|---|---|---|---|
| **Content** | 100 m resolution elevation information of the AP (Graham Land, 63–70° S) | Inventory of 1590 glacier basins of the AP (Graham Land, 63–70° S) on the mainland and surrounding islands | New rock outcrop dataset for Antarctica | bedrock data set for the AP (Graham Land, 63–70° S) on a 100m grid | Drainage divides of the Antarctic ice sheet |
| **Sources** | ASTER Global Digital Elevation Model (GDEM) | DEM of Cook et al. (2012), LIMA (Bindschadler et al., 2008), Grounding line based on the Antarctic Surface Accumulation and Ice Discharge (ASAID) project data source (Bindschadler et al., 2011) | Landsat 8 data | Simple ice dynamic modelling with a variety of available datasets (surface mass balance, point ice thickness and ice flow velocity) | GLAS/ICESat 500 m laser altimetry DEM (DiMarzio, 2007), Landsat Image Mosaic of Antarctica (Bindschadler et al., 2008) and the MODIS Mosaic of Antarctica (Haran et al., 2005) |
| **Access** | http://nsidc.org/data /nsidc-0516.html | http://add.scar.org/ (available only with a limited number of attributes) | http://add.scar.org/ | Available online from the article's supplement (doi:10.5194/tc-8-1261-2014-supplement) | http://icesat4.gsfc.n asa.gov/cryo_data/ ant_grn_drainage_s ystems.php |
| **Reference** | Cook et al. (2012) | Cook et al. (2014) | Burton-Johnson et al. (2016) | Huss and Farinotti (2014) | Zwally et al. (2012) |
| **Application in this study** | Calculation of (a) glacier specific topographic parameters (min, max., mean, median elevation, slope, aspect), (b) overall and glacier specific hypsometry, and (c) thickness grid combined with the bedrock elevation grid of Huss and Farinotti (2014) | Initial dataset for the generation of glacier outlines | Used to remove the (ice free) rock outcrops from the glacier catchment outlines to generate glacier outlines | Calculation of the thickness grid combined with the DEM of Cook et al. (2012) | Separation of the glaciers form the ice sheet |

**Table 2. Glacier parameters in the attribute table of the inventory of the AP.**

| Name | Item | Glacier example | Description |
|---|---|---|---|
| Name | Name | Romulus Glacier | String, partially available |
| Satellite Image Date | SI_DATE | 19.02.2001 | Date of the Satellite image used for digitizing |
| Year | SI_YEAR | 2001 | Year the outline is representing |
| Satellite Image Type | SI_TYPE | Landsat7 | Instrument name e.g. Landsat 7 |
| Satellite Image ID | SI_ID | LE7220108000105050 | Original ID of image |
| Coordinates | Lat, long | -68.391218, -66.82767 | Decimal degree |
| Primary classification | Class | 6 (mountain glacier) | cf. Cook et al. (2014) |
| Form | Form | 2 (compound basin) | cf. Cook et al. (2014) |
| Front | Front | 4 (calving) | cf. Cook et al. (2014) |



| Confidence | Confidence | 1 Confident about all (Class, Form and Front) classification types | cf. Cook et al. (2014) |
|---|---|---|---|
| Mainland/island | Mainl_Isl | 1 (situated on mainland) | cf. Cook et al. (2014) |
| Area | area | 68.911 km$^2$ | km$^2$ |
| Connectivity level | CL | 1 (weak connection) | cf. Sect. 4.2 |
| Sector | Sector | SW | NW, NE, SW or SE |
| Minimum elevation | min_elev | 4.636 m a.s.l. | m a.s.l. |
| Maximum elevation | max_elevation | 1610.625 m a.s.l. | m a.s.l. |
| Mean elevation | mean_elev | 466.465 m a.s.l. | m a.s.l. |
| Median elevation | med_elev | 425.577 m a.s.l. | m a.s.l. |
| Mean Aspect in degree | mean_asp_d | 222.605 ° | ° |
| Mean Aspect nominal | mean_aspect | SW | 8 cardinal directions |
| Aspect Sector | asp_sector | 6 | Clockwise numbering of the 8 cardinal directions |
| Mean slope | mean_slope | 13.891 ° | ° |
| Total volume | tot_vol | 13.417 km$^3$ | km$^3$ |
| Volume below sea level | vol_below | 7.985 km$^3$ | km$^3$ |
| Mean thickness | mean_thick | 191.427 m | m |

**Table 3. Glacier number, area, volume, volume grounded below sea level, the corresponding percentages and SLE per sector. For the estimation of SLE see Sect. 4.3.**

| Sector | Count | Count with volume info | Area [km$^2$] | Count [%] | Area [%] | Volume [km$^3$] | Volume [%] | Volume$_{<0}$ [km$^3$] | Volume$_{<0}$ [%] | SLE [mm] |
|---|---|---|---|---|---|---|---|---|---|---|
| NW | 704 | 679 | 17 218 | 44 | 18 | 4 026 | 12 | 1 093 | 27 | 7 |
| NE | 246 | 237 | 18 278 | 15 | 19 | 6 133 | 18 | 2 939 | 48 | 7 |
| SW | 378 | 362 | 31 130 | 24 | 33 | 13 365 | 39 | 4 849 | 36 | 20 |
| SE | 261 | 261 | 28 647 | 17 | 30 | 11 065 | 32 | 2 890 | 26 | 20 |
| **Total** | 1 589 | 1 539 | 95 273 | 100 | 100 | 34 590 | 100 | 11 771 | 100 | 54 |

**Figures**

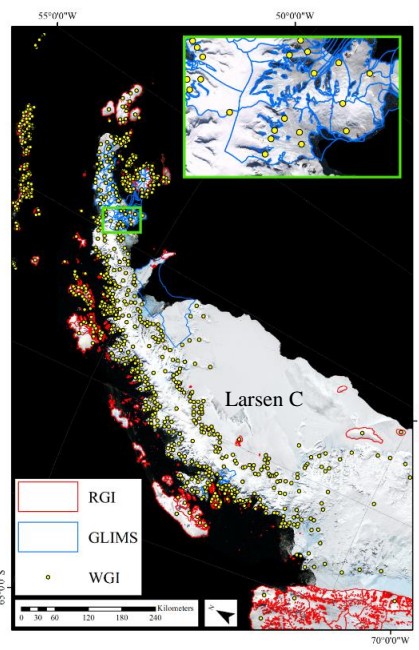


**Figure 1. LIMA overlaid by existing GLIMS and RGI glacier outlines and WGI glacier point locations for Graham Land on the AP. Inset map illustrating that the distribution of the WGI points does not enable assignation of points to individual glacier outlines.**

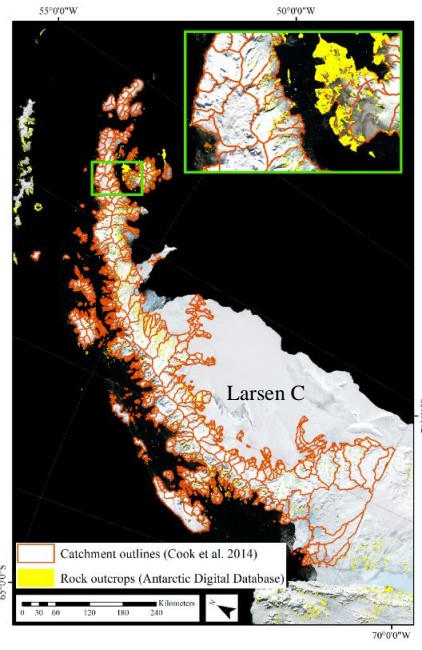

**Figure 2. Glacier catchment outlines of Cook et al. (2014) and the newest rock outcrops dataset from Burton-Johnson et al. (2016)**
**overlaying the LIMA.**



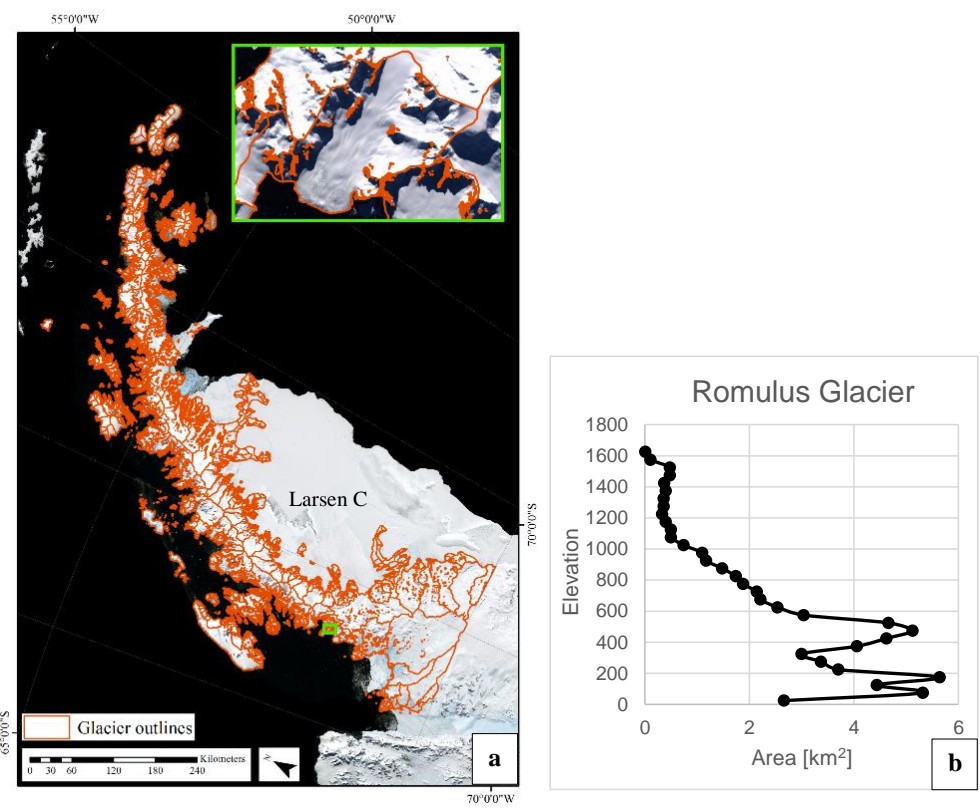

**Figure 3. a) Glacier outlines. Inset map showing Romulus glacier referred to in Table 2 and b) exemplifying the glacier specific hypsometry.**

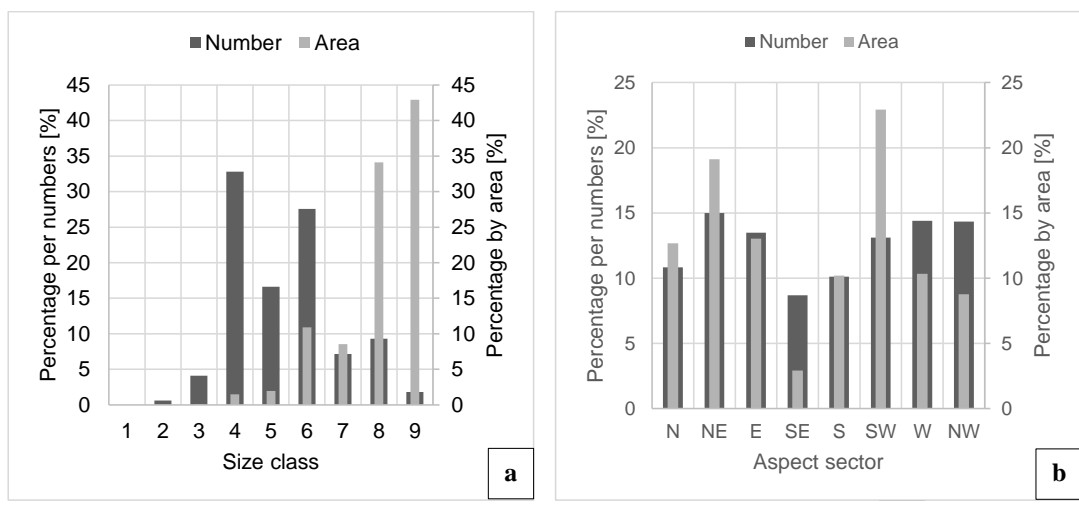

**Figure 4. a) Percentage of glacier count and area per size class and b) percentage of glacier count and area per aspect sector.**



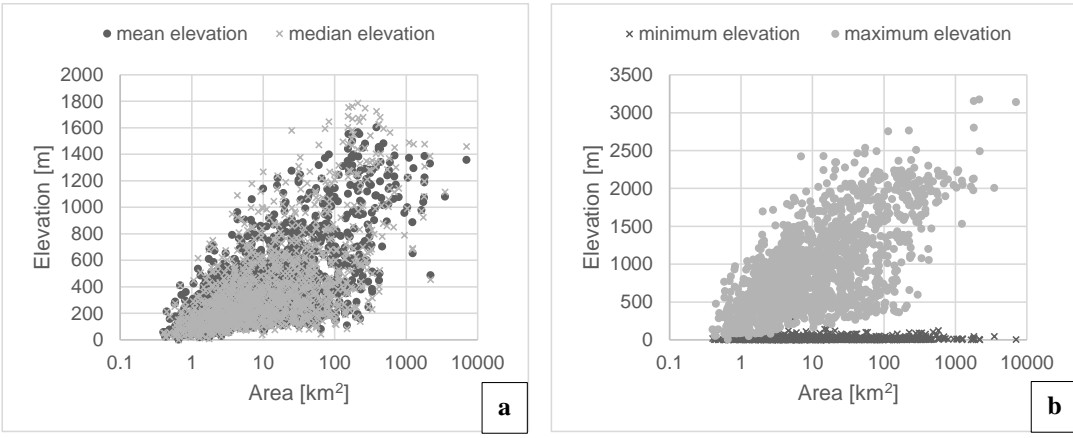

**Figure 5. a) Mean and median elevation vs. area and b) minimum and maximum elevation vs. area of the 1541 glaciers including elevation information.**

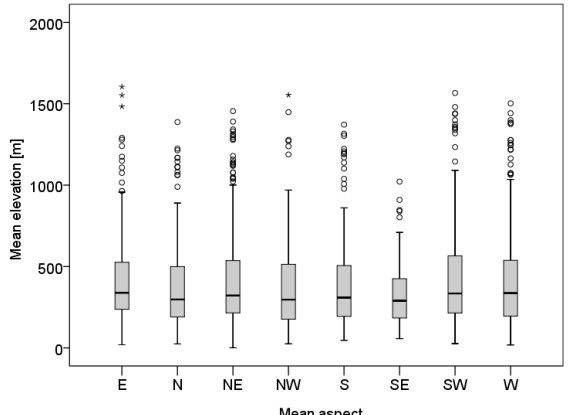

**Figure 6. Mean glacier elevation vs. mean glacier aspect of 1541 glaciers.**

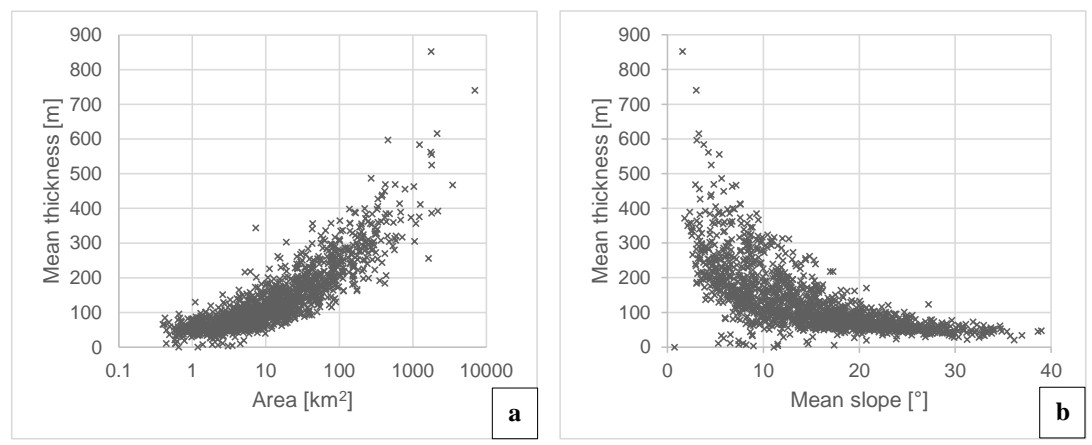

**Figure 7. Scatter plot of the 1541 glaciers involving thickness information. a) Mean thickness vs. area and b) mean thickness vs. mean slope.**



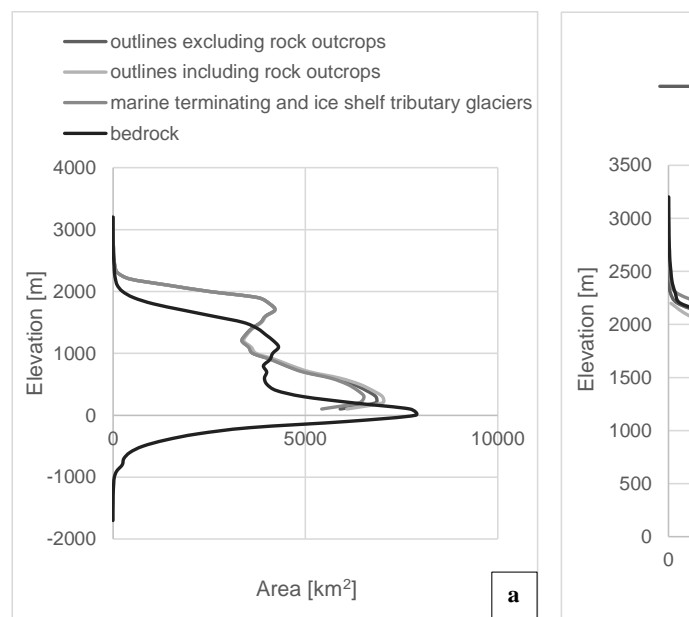
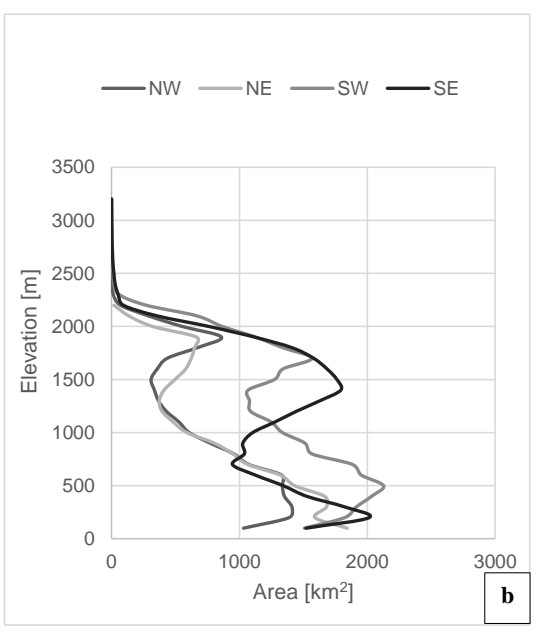

**Figure. 8. Glacier hypsometry of the total area covered by the DEM. a) Total areal distribution excluding rock outcrops, the areal distribution including rock outcrops, areal distribution for marine-terminating and ice shelf nourishing glaciers and areal distribution of the underlying bedrock. b) Areal distribution of the glacier cover per sector.**

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
