# Peer review of "A complete glacier inventory of the Antarctic Peninsula based on Landsat7 images from 2000-2002 and other pre-existing datasets"

_Earth System Science Data, 2016_

## Referee Comment (RC1) · B. Marzeion (Referee) · 19 Oct 2016

Huber et al. present a complete and comprehensive inventory of glaciers on the Antarctic Peninsula, based on the aggregation of a diverse set of previously published data. The availability of such data sets is a prerequisite for understanding, and quantifying, the sensitivity and response of glaciers to climate variability and change, including all the consequences (in the case of the AP, primarily sea-level rise). A data set as presented here is long-needed for the AP, is a great step towards completion of such inventories on the global scale and will be used in many future studies.

The manuscript is generally well written (see below for a few specific comments), and the authors describe the workflow well and understandable also for non-specialist

readers. However, before I can recommend the manuscript for publication, one general (and somewhat major) and several specific (and mostly minor) comments should be addressed. Please see below for the details.

**General comments:**

- I see only one issue with the manuscript that will require some more substantial work to address: the handling (or lack of handling) of uncertainties. The authors address uncertainties in section 6.3. However, they do not attempt to translate the discussion here into uncertainties regarding their results - e.g., the hard-to-determine grounding line (Sect. 6.3.3) has a strong influence on the glacier area: how does that translate into uncertainty of the total area number for the AP given in the abstract and elsewhere?
  Addressing this properly, i.e. trying to come up with uncertainty estimates for the numbers given in the abstract, the main text, and Table 3, will also require a reorganization of the manuscript. What is now in Sect. 6.3 would have to be discussed (together with the derivation of the uncertainty estimates) before the results section (Sect. 5).
  It may not be possible to estimate the uncertainties for all the derived variables, and for some where it is possible, it may only be very indicative (e.g., is the error in the 1 % range or in the 10 % range). But in the current shape, the manuscript creates the (wrong, I'm sure) impression of blindness of the authors towards this issue – also because of the way many numbers are presented (e.g., mean thickness including digits on the mm scale, or the total area of glaciers on the AP to the 0.1 km2 scale).

**Specific comments:**

- L20ff: all the numbers given in the abstract should include an uncertainty estimate

– see above.

- L26ff: since you do not discuss the ELA in the manuscript, having point (1) in the abstract is a bit of a surprise without additional information, I would recommend removing it; points (2) and (3) could be combined into one statement on the sensitivity of the region to marine-induced ice dynamic effects.

- L39/Fig. 1: the first reference to Fig. 1 comes before the definition of the abbreviation LIMA - so this definition should perhaps be in the caption of Fig. 1.

- L65: "the part of the AP north of 70 S represents..."

- L78: "experience" is typeset is a smaller font?

- L79: delete "help".

- Sect. 3: it would help in some places to change the formulation to better separate what was done by the authors of this manuscript from what was done by the authors of the references. E.g., instead of writing ".. groundling lines ... were used, modified in places..." (L119) an active voice clarifies: "... they used grounding lines and modified them in places..."; or instead of "Other sources ... were used" (L164): "They used other sources...".

- L107: it would be good if you indicate (one sentence) how Cook et al. (2012) deal with these gaps.

- L188: "'...glaciers on Greenland with a strong dynamic connection the Greenland ice sheet should..."

- L190: "strong connection" – I think "strong" is ambiguous here (especially in connection with the word "strong" in the previous sentence. As it refers to space here, perhaps "broad" or "extended" would be better?

- L207ff: I think the greater problem (still not very big...) than the spatial inconsistency is that the thickness estimation also depends on velocity, which is of course connected to ice divides, outcrops, etc. That is, there is a certain degree of internal inconsistency among the data sets used. This could only be fixed by repeating (perhaps even iteratively) certain parts of the analyses that lead to the data sets used – this would obviously be too much to ask for, but I think this problem needs to be mentioned in the manuscript, probably best in the discussion section.

- L219ff: can be shortened to "The sea level equivalent (SLE) of the ice volume is calculated by assuming a mean ice density of 900 kg m-3 and dividing it by the ocean surface area (3.625 × 108 km2; Cogley, 2012)."

- L225: isostatic effects (increasing ocean volume and lifting the land surface) probably have a bigger effect than the dilution, cooling, and ocean dynamics.

- L230: "100 m bins"

- L238f: isn't this the same as stating "the rock outcrops cover an area of 1709 km2"? - it would be simpler.

- L240: giving the area of the smallest glacier in the data set has limited value, as it is the smallest "detected" glacier, and there are presumably a large number of very small glaciers missing in the data set (see the discussion in Pfeffer et al., 2014, around that). It would be more helpful to refer to the histogram (Fig. 4).

- Table 2: remove digits in the numbers that are clearly within the uncertainty range (e.g., giving minimum elevation on a mm scale...).

- L247: "a csv-file".

- L259/Fig. 4: please specify the size classes (either in a small table, or perhaps better, on the figure axis).

- L268: "...being 300 m or more higher than ..."

- L269: what does "in the south" mean here?

- Tab. 3: it would be good to have uncertainty estimates also here.

- Fig. 6 (also Fig. S3): please indicate the percentiles corresponding to the box and whiskers in the caption.

- L399: "For instance, for a glacier with missing values at the coast, the..."

- L405f: this would also help removing internal inconsistencies, see above.

- L435: "...of a land-terminated glacier...".

- L459: repetition from the paragraph above.

- L470f: "The lowest 800 m and hence 50 % of the glacierized area are prone to rising ablation and mass loss,..." I don't understand where the number 800 m comes from?

- L238/L481: excluding islands $< 0.5$ km2 or 0.05 km2?

---

## Referee Comment (RC2) · Anonymous Referee #2 · 31 Oct 2016

General

In this manuscript the authors present and describe a very nice dataset as the so far most complete glacier inventory of the Antarctic Peninsula. This is very useful and will be used as baseline data in coming assessments of the sensitivity and responses of AP glaciers to climate changes, based on monitoring, modelling and remote sensing. The paper is in general very well written, clear and easy to follow. I am not able to suggest any improvements of the language. It is nice to see that the inventory is easy accessible at the GLIMS website. The paper can be published nearly as it is but the authors should go through the paper and clear up the more specific comments I have below.

[Figure]

Specific comments.

It is nice that they have a fairly large section 6 on Uncertainties, but still I think they could have done more on that. Especially the volume estimates should have been given with error bars or +-. There must be fairly large uncertainties in ice thickness estimates and thus also in the volume estimates. In Table 3 they give exact numbers on volumes in each sector, but these estimates must have a fairly large uncertainty. Also in Table 2 were they give an example of a glacier in the inventory they give very exact numbers. How is the position of the point where coordinates are taken selected ? They give the position with six decimals. Thus it is a point derived from a GIS tool, but still the location selected must follow a definition. In the same Table 2, and I suppose thus for all glaciers in the inventory, they give area with three decimals (68.911 km2) and the same with elevations given on mm scale. This does not make sense to me. Mean thickness is given as 191.427 m so again on mm scale. Thus error bars/intervals would be informative together With s short comment on the uncertainties.

In line 123 they write 1590 glacier catchments while in line 209 it says 1589 In line 292 it says that the volume of AP glaciers is 33 770 km3, but in Table 4 it says 34 590 km3. The total glacier area is given as different numbers: in line 123 it says 96 982 km2, in line 238 it says 95 273.2 This number 95 273 is also listed in Table 3 and given as the total area in the conclusions in line 481, in line 323 it says 93 767 km2, in line 358 it says 94 743 km2. They should go through the numbers so it is consistent throughout.

In line 387 they refer to Supplement 5. But there is no Supplement 5 in the file I can find. The figures with map plots, Figs.1, 2, 3 and also Supplement 4 are all nice and give some good information, but in a printed version they will be almost impossible to read. I had to zoom in my pdf-file to 300 %. Then it is fine. Most users of the inventory will work on digital versions so maybe this is fine.

The other figures present data from statistics and are to some extent informative but give few surprises. It is fairly obvious and as expected that in general larger areas give
larger thickness, steep glaciers are thinner than less steep glaciers, and thus also that large glaciers in general has a lower slope than small.

Fig. 8. a. This figure is reproduced in grey-scale. It is almost impossible to separate out the information between the three categories; 1) outlines excluding rock outcrops, 2) outlines including rock outcrops and 3) marine terminating and ice shelf tributary glaciers. It is only in the elevation range 200-500 meter that there is any notable difference. Otherwise they overlap. I suggest that they only show one curve for outlines excluding rock outcrops in addition to the bedrock curve.

---

## Author Comment (AC1) · 7 Dec 2016

Author response on:
Huber et al. present a complete and comprehensive inventory of glaciers on the Antarctic Peninsula, based on the aggregation of a diverse set of previously published data.

[Figure]

The availability of such data sets is a prerequisite for understanding, and quantifying, the sensitivity and response of glaciers to climate variability and change, including all the consequences (in the case of the AP, primarily sea-level rise). A data set as presented here is long-needed for the AP, is a great step towards completion of such inventories on the global scale and will be used in many future studies. The manuscript is generally well written (see below for a few specific comments), and the authors describe the workflow well and understandable also for non-specialist readers. However, before I can recommend the manuscript for publication, one general (and somewhat major) and several specific (and mostly minor) comments should be addressed. Please see below for the details.

General comments:  ĺ I see only one issue with the manuscript that will require some more substantial work to address: the handling (or lack of handling) of uncertainties. The authors address uncertainties in section 6.3. However, they do not attempt to translate the discussion here into uncertainties regarding their results - e.g., the hard-to determine grounding line (Sect. 6.3.3) has a strong influence on the glacier area: how does that translate into uncertainty of the total area number for the AP given in the abstract and elsewhere? Addressing this properly, i.e. trying to come up with uncertainty estimates for the numbers given in the abstract, the main text, and Table 3, will also require a reorganization of the manuscript. What is now in Sect. 6.3 would have to be discussed (together with the derivation of the uncertainty estimates) before the results section (Sect. 5). It may not be possible to estimate the uncertainties for all the derived variables, and for some where it is possible, it may only be very indicative (e.g., is the error in the 1 % range or in the 10 % range). But in the current shape, the manuscript creates the (wrong, I'm sure) impression of blindness of the authors towards this issue – also because of the way many numbers are presented (e.g., mean thickness including digits on the mm scale, or the total area of glaciers on the AP to the 0.1 km2 scale).

Response: We agree, that specific uncertainty estimations are missing and have thus

added the estimates from the related studies. We are fully aware that uncertainties in glacier area (e.g. due to uncertainties in the position of grounding lines or drainage divides) impact on other parameters such as hypsography, topographic indices and glacier volume. In a similar way uncertainties of the DEM propagate into these parameters as well as into the derived area. As the impact of these multiple-dependencies are difficult to determine (has it ever been done?) and might thus give a nice study on its own (requiring to manipulate the original datasets), we prefer here to just report the uncertainties for the individual datasets as given in the related studies. To better illustrate the complexity of the issue for the reader, we have re-structured the section and added a systematic description of the mutual dependencies.

Regarding the decimals: We are aware, especially considering the uncertainties, that giving three decimals might be seen as inappropriate and have reduced the decimals to one digit (this is also required as the smallest glacier in the sample has a size of 0.3 km2). However, to retain full traceability with the original datasets and allow comparisons to other studies, we would prefer to stay with this precision rather than rounding values to the appropriate uncertainty. We hope that together with the now added more detailed and systematic description of uncertainties the reader can interpret the given numbers correctly.

Specific comments: • L20ff: all the numbers given in the abstract should include an uncertainty estimate

Response: See above, these are difficult to determine correctly. (We have thus used standard values from the literature to give at least an indication).

• L26ff: since you do not discuss the ELA in the manuscript, having point (1) in the abstract is a bit of a surprise without additional information, I would recommend removing it; points (2) and (3) could be combined into one statement on the sensitivity of the region to marine-induced ice dynamic effects.

Response: We discuss the ELA in Section 6.2 and 6.4. Also points (2) and (3) are

discussed separately. Therefore, we would like to keep these three points.

• L39/Fig. 1: the first reference to Fig. 1 comes before the definition of the abbreviation LIMA - so this definition should perhaps be in the caption of Fig. 1.

Response: Done.

• L65: "the part of the AP north of 70 S represents..."

Response: Done.

• L78: "experience" is typeset is a smaller font?

Response: Done.

• L79: delete "help".

Response: Done.

• Sect. 3: it would help in some places to change the formulation to better separate what was done by the authors of this manuscript from what was done by the authors of the references. E.g., instead of writing ".. groundling lines ... were used, modified in places..." (L119) an active voice clarifies: "... they used grounding lines and modified them in places..."; or instead of "Other sources ... were used" (L164): "They used other sources...".

Response: Done.

• L107: it would be good if you indicate (one sentence) how Cook et al. (2012) deal with these gaps.

Response: Cook et al. (2012) do not further comment or deal with these gaps. They just note that they exist ("Some anomalies along the coast have been removed, resulting in small gaps, and the DEM has a small number of remaining artefacts."). As our intention here was bringing the different datasets together as is (for traceability) rather than manipulating them for improved results, we have not further corrected them. However, we fully agree that this could be a next issue to look at in future study.

• L188: "'...glaciers on Greenland with a strong dynamic connection the Greenland ice sheet should..."

Response: Done.

• L190: "strong connection" – I think "strong" is ambiguous here (especially in connection with the word "strong" in the previous sentence. As it refers to space here, perhaps "broad" or "extended" would be better?

Response: Done.

• L207ff: I think the greater problem (still not very big...) than the spatial inconsistency is that the thickness estimation also depends on velocity, which is of course connected to ice divides, outcrops, etc. That is, there is a certain degree of internal inconsistency among the data sets used. This could only be fixed by repeating (perhaps even iteratively) certain parts of the analyses that lead to the data sets used – this would obviously be too much to ask for, but I think this problem needs to be mentioned in the manuscript, probably best in the discussion section.

Response: We agree and have now described the problem in Section 6.3.1 in detail. There is indeed little we can do about it now, but we hope that once more complete and accurate information becomes available (e.g. better DEMs, complete flow velocity fields) calculations will be redone by the respective authors.

• L219ff: can be shortened to "The sea level equivalent (SLE) of the ice volume is calculated by assuming a mean ice density of 900 kg m-3 and dividing it by the ocean surface area (3.625 _ 108 km2; Cogley, 2012)."

Response: Done.

• L225: isostatic effects (increasing ocean volume and lifting the land surface) probably have a bigger effect than the dilution, cooling, and ocean dynamics.

[Figure]

Response: We agree and mention now, that also the isostatic effect is not taken into account.

• L230: "100 m bins"

Response: Done.

• L238f: isn't this the same as stating "the rock outcrops cover an area of 1709 km2"? - it would be simpler.

Response: Done.

• L240: giving the area of the smallest glacier in the data set has limited value, as it is the smallest "detected" glacier, and there are presumably a large number of very small glaciers missing in the data set (see the discussion in Pfeffer et al., 2014, around that). It would be more helpful to refer to the histogram (Fig. 4).

Response: We agree and have deleted the sentence about smallest and largest glaciers. Later in the paragraph we already refer to Figure 4.

• Table 2: remove digits in the numbers that are clearly within the uncertainty range (e.g., giving minimum elevation on a mm scale...).

Response: Done.

• L247: "a csv-file".

Response: Done.

• L259/Fig. 4: please specify the size classes (either in a small table, or perhaps better, on the figure axis).

Response: Done on the figure axis.

• L268: "...being 300 m or more higher than ..."

Response: Done.

• L269: what does "in the south" mean here?

Response: Clarified: "The highest elevation is in southern Graham Land. . ."

• Tab. 3: it would be good to have uncertainty estimates also here.

Response: See above.

• Fig. 6 (also Fig. S3): please indicate the percentiles corresponding to the box and whiskers in the caption.

Response: Done.

• L399: "For instance, for a glacier with missing values at the coast, the..."

Response: Done.

• L405f: this would also help removing internal inconsistencies, see above.

Response: See above.

• L435: "...of a land-terminated glacier..."

Response: Done.

• L459: repetition from the paragraph above.

Response: Done. Removed.

• L470f: "The lowest 800 m and hence 50 % of the glacierized area are prone to rising ablation and mass loss,..." I don't understand where the number 800 m comes from?

Response: Above we mention that the hypsometry reveals the half of the glacierized areas being situated below 800 m a.s.l. We refer to that. For clarification we have now written: "With respect to the hypsometry, the lowest 800m and hence. . ."

• L238/L481: excluding islands < 0.5 km2 or 0.05 km2?

[Figure]

Response: Done. (0.5 km2).

---

## Author Comment (AC2) · 7 Dec 2016

Author response on: Interactive comment on "A complete glacier inventory of the Antarctic Peninsula based on Landsat7 images from 2000–2002 and other pre-existing datasets" by Jacqueline Huber et al.

Anonymous Referee #2

General: In this manuscript the authors present and describe a very nice dataset as the so far most complete glacier inventory of the Antarctic Peninsula. This is very useful and will be used as baseline data in coming assessments of the sensitivity and responses of AP glaciers to climate changes, based on monitoring, modelling and

remote sensing. The paper is in general very well written, clear and easy to follow. I am not able to suggest any improvements of the language. It is nice to see that the inventory is easy accessible at the GLIMS website. The paper can be published nearly as it is but the authors should go through the paper and clear up the more specific comments I have below.

Specific comments: It is nice that they have a fairly large section 6 on Uncertainties, but still I think they could have done more on that. Especially the volume estimates should have been given with error bars or +-. There must be fairly large uncertainties in ice thickness estimates and thus also in the volume estimates. In Table 3 they give exact numbers on volumes in each sector, but these estimates must have a fairly large uncertainty. Also in Table 2 were they give an example of a glacier in the inventory they give very exact numbers. How is the position of the point where coordinates are taken selected? They give the position with six decimals. Thus it is a point derived from a GIS tool, but still the location selected must follow a definition. In the same Table 2, and I suppose thus for all glaciers in the inventory, they give area with three decimals (68.911 km2) and the same with elevations given on mm scale. This does not make sense to me. Mean thickness is given as 191.427 m so again on mm scale. Thus error bars/intervals would be informative together With s short comment on the uncertainties.

Response: We fully agree that providing three decimals is of limited value when considering the uncertainties. In particular, for the thickness estimates the mm scale is certainly meaningless. For normal we give two decimals for area and one for thickness and have thus now reduced both to one decimal. We also added an estimation of uncertainty to the respective numbers where possible. As uncertainties depend in a complex way on each other (e.g. area on the position of drainage divides as derived from the DEM) we have also added a section describing these dependencies in a systematic way to inform about error propagation at least in a qualitative way. We further added the various uncertainty estimates from the original studies to also provide

quantitative values. Further reassessment

Regarding the selection and precision of the label point for each glacier we have now clarified that the points were either selected manually following the guidelines of the GLIMS Analysis Tutorial (Raup and Khalsa, 2007) or assigned automatically by the GIS within the glacier polygon. Its value must have at least 4 decimals in geographic coordinates to have a 10 m precision in metric coordinates. Assuming they should be accurate on the metre, 5 decimals are required. The given six decimals are thus too precise but have been retained for computational reasons (e.g. floating point values are stored in the GIS with double precision).

Comment: In line 123 they write 1590 glacier catchments while in line 209 it says 1589 In line 292 it says that the volume of AP glaciers is 33 770 km3, but in Table 4 it says 34 590 km3. The total glacier area is given as different numbers: in line 123 it says 96 982 km2, in line 238 it says 95 273.2. This number 95 273 is also listed in Table 3 and given as the total area in the conclusions in line 481, in line 323 it says 93 767 km2, in line 358 it says 94 743 km2. They should go through the numbers so it is consistent throughout.

Response: Thank you for recognizing the differences. Regarding the volume in line 292 and the area in line 358 we indeed provided the wrong values. The correct ones are 34 590 km3 for the volume and 95 273.2 km2 for the area. Both have been corrected now. Regarding the other numbers of glacier catchment area, glacier area and glacier number: The numbers are correct as they refer to different datasets (with very subtle differences): First, there are 1590 glacier catchments with an area of 96 982 km2, after removing the rock outcrops, we have 1589 glaciers with an area of 95 273.2 km2. Finally, the 93 767km2 refer to the area covered by the DEM (not all glaciers are covered by the DEM).

Comment: In line 387 they refer to Supplement 5. But there is no Supplement 5 in the file I can find. The figures with map plots, Figs.1, 2, 3 and also Supplement 4 are
all nice and give some good information, but in a printed version they will be almost impossible to read. I had to zoom in my pdf-file to 300 %. Then it is fine. Most users of the inventory will work on digital versions so maybe this is fine.

Response: We hope that the PDF quality and size of the figures stays like this, so that a 300% zoom reveals all details. For the publication, the figures should only be an illustration of the various datasets and full details are accessible from the freely available digital data.

Comment: The other figures present data from statistics and are to some extent informative but give few surprises. It is fairly obvious and as expected that in general larger areas give larger thickness, steep glaciers are thinner than less steep glaciers, and thus also that large glaciers in general has a lower slope than small.

Response: We agree that some statistics give few surprises as they might only confirm well-known glaciological / physical relationships. However, we would like to provide them nevertheless as the glaciers in this region are quite different from other regions and the figures give quick access to key characteristics of the new dataset. As such statistics are also used in most other studies to characterize the glaciers/inventory they also allow easy comparison across regions.

Comment: Fig. 8. a. This figure is reproduced in grey-scale. It is almost impossible to separate out the information between the three categories; 1) outlines excluding rock outcrops, 2) outlines including rock outcrops and 3) marine terminating and ice shelf tributary glaciers. It is only in the elevation range 200-500 meter that there is any notable difference. Otherwise they overlap. I suggest that they only show one curve for outlines excluding rock outcrops in addition to the bedrock curve.

Response: We fully agree that the different lines are difficult to distinguish and have adjusted line colours and styles to be more clear. The curve "outlines including rock outcrops" has been removed. The curve "marine terminating and ice shelf tributary glaciers" is still shown as we refer to it in the text.